# Periodontal Indices as Predictors of Cognitive Decline: Insights from the PerioMind Colombia Cohort

**DOI:** 10.3390/biomedicines13010205

**Published:** 2025-01-15

**Authors:** Catalina Arévalo-Caro, Diego López, Jose Antonio Sánchez Milán, Cristina Lorca, María Mulet, Humberto Arboleda, Sergio Losada Amaya, Aida Serra, Xavier Gallart-Palau

**Affiliations:** 1+Pec Proteomics Research Group (+PPRG), Neuroscience Area, Biomedical Research Institute of Lleida Dr. Pifarré Foundation (IRBLLEIDA), University Hospital Arnau de Vilanova (HUAV), 25198 Lleida, Spain; cmarevaloc@unal.edu.co (C.A.-C.);; 2+Pec Proteomics Research Group (+PPRG), Department of Medical Basic Sciences, Faculty of Medicine, University of Lleida (UdL), 25198 Lleida, Spain; 3Grupo de Investigación en Periodoncia y Medicina Periodontal, Departamento de Ciencias Básicas y Medicina Oral, Facultad de Odontología, Universidad Nacional de Colombia, Sede Bogotá, Carrera 30 No. 45-03, Edificio 210, Bogotá 11001, Colombia; 4Neuroscience and Cell Death Research Groups, Medical School and Genetic Institute, Universidad Nacional de Colombia, Sede Bogotá, Carrera 30 No. 45-03, Bogotá 111321, Colombia

**Keywords:** dentistry, periodontitis, cognitive decline, mild cognitive impairment, oral health, prodromal Alzheimer’s disease, risk factors, predictive models, cohort studies, periodontal indices

## Abstract

**Background**: Poor oral health and periodontitis have been epidemiologically linked to cognitive decline and mild cognitive impairment (MCI) in older adults. However, specific metrics directly linking these clinical signs are exceedingly limited. **Methods**: To address this gap and develop novel tools to help clinicians identify individuals at risk of cognitive decline, we established the PerioMind Colombia Cohort, comprising elderly Colombian subjects who underwent comprehensive neurocognitive and periodontal evaluations. **Results**: The results revealed that subjects diagnosed with MCI exhibited significantly higher scores in specific periodontal indices, including gingival erythema and pocket depth parameters. The predictive model identified positive associations with MCI, with gingival erythema showing the strongest correlation, followed by the presence of periodontitis and variations in pocket depth measurements. Additionally, lower educational attainment was associated with a higher likelihood of being classified in the periodontitis-MCI group. **Conclusions**: Here, we show that specific altered periodontal metrics are associated with MCI diagnosis, and the generated results provide defined metric ranges for identifying individuals at risk. Upon validation in larger cohorts, the findings reported here could offer dental practitioners and clinicians innovative tools to identify individuals at risk of MCI and age-related dementias through routine oral health assessments, thereby enabling more accessible and highly sought-after early intervention strategies in both developing and developed countries.

## 1. Introduction

The global prevalence of dementia was estimated at approximately 55 million individuals in 2021, with projections indicating an increase to 78 million by 2030 and 139 million by 2050 [1]. This substantial rise is primarily driven by population aging, particularly in low- and middle-income countries, which currently account for approximately 60% of individuals living with dementia [2]. In Latin America, the prevalence of aging-associated dementias is estimated to range between 5% and 10% among individuals over 60 years of age, with projections indicating a significant upward trend consistent with global patterns [3,4,5]. Recent studies emphasize the significance of conducting research within this specific elderly population, as misdiagnosis and underdiagnosis remain prevalent [6]. These challenges are primarily attributed to limited access to culturally appropriate diagnostic tools and persistent healthcare disparities across countries in the region [7], as well as the influence of the mestizo population’s genetic diversity, which complicates dementia characterization [7]. Similarly, most research is based on European ancestry, potentially leading to biases in understanding and treating dementia in mestizo populations [7].

Colombia presents an optimal setting for investigating mestizo populations within Latin America owing to its heterogeneous genetic and cultural heritage, which exemplifies the broader demographic patterns of the region [7]. Furthermore, Colombia’s increasing emphasis on public health research, in conjunction with its healthcare challenges, renders it a significant context for examining global health disparities and population dynamics [6]. Dementia prevalence in Colombia ranges from 5% to 7% in individuals over 60 years of age, aligning with the aforementioned broader Latin American trends [6]. However, specific geographical areas within the country have exhibited an exceptionally high prevalence of 23.6% in individuals over 60 years old, attributed to socioeconomic factors and poorly defined chronic comorbidities [8].

Mild cognitive impairment (MCI) is an early stage of cognitive decline [9,10,11,12], characterized by noticeable but mild impairments in memory, language, or spatial perception, while daily tasks remain largely unaffected [12]. Individuals with MCI exhibit an elevated risk of progressing to dementia, with lifestyle factors, genetic predispositions, and comorbidities playing pivotal roles in this progression [13,14]. Therefore, the identification of MCI in elderly populations is crucial for decelerating the progression of this debilitating and fatal syndrome, thereby facilitating timely therapeutic interventions to enhance patient outcomes and address healthcare challenges associated with aging populations [15].

Conversely, periodontitis is a progressive inflammatory disease characterized by the destruction of the soft and hard tissues of the periodontal complex, driven by interactions between dysbiotic microbial communities and aberrant immune responses [16]. A critical determinant of periodontitis development and progression is the elevated concentration of pathogenic bacteria within the dental biofilm, which elicits a detrimental immune response [17,18]. Notably, the deleterious effects of periodontitis extend beyond the oral cavity and have been associated with systemic health issues, as we and other colleagues have previously elucidated [19,20,21]. Furthermore, this disease appears to be a chronic comorbidity commonly observed in individuals affected by aging-related dementias [22]. Emerging epidemiological data from various countries suggest that periodontitis may play a significant role in influencing the onset and progression of age-related dementias [23]. This association is likely attributable to the systemic inflammation and immune responses elicited by chronic periodontal infections, which could exacerbate neuroinflammatory processes linked to cognitive decline [23]. Additionally, the prevalence of periodontitis in aging populations underscores its potential as a modifiable risk factor for dementia [22]. Elucidating this connection could facilitate the development of targeted preventive strategies to mitigate dementia’s burden in susceptible populations.

Considering these factors, this study evaluates the potential interactions between MCI and periodontitis-associated pathogenic factors in elderly individuals from a mestizo population. To accomplish this objective, we established the PerioMind Colombia Cohort, conducting comprehensive assessments of both neurocognitive and periodontal parameters associated with MCI and periodontitis. Our findings reveal significant and specific relationships between periodontal disease and MCI, providing insights into novel diagnostic tools. By identifying key periodontal metrics strongly associated with MCI, we propose that routine oral examinations, which are more readily available than specialized neurocognitive assessments in Colombia and internationally, could serve as an effective method for identifying individuals at risk of cognitive decline. Therefore, this study aims to address a significant gap in understanding the relationship between oral and cognitive health by providing novel data and diagnostic tools that advance early intervention strategies for aging populations globally.

## 2. Materials and Methods

### 2.1. Study Design and Participants

The establishment of novel cohorts to investigate diverse and specific aspects of MCI in elderly populations is imperative for advancing dementia prevention strategies [24]. Therefore, as previously referred [25], cohort-based studies provide valuable insights into region-specific challenges, facilitating the development of culturally tailored healthcare interventions.

A cross-sectional observational study was designed and conducted at the Institute of Genetics, Universidad Nacional de Colombia. This research design was selected due to its suitability for exploratory investigations aimed at identifying specific clinical associations among variables, as previously delineated [26]. This approach allows for the efficient collection of representative data within resource constraints, providing preliminary evidence to guide future longitudinal studies for causal confirmation [26]. Participant recruitment occurred from January to December 2023. The study enrolled a total of 70 participants. The cohort comprised Colombian men and women, predominantly residents of Bogotá, aged 65 years and older. All participants provided informed consent prior to study inclusion. Subsequently, subjects underwent comprehensive neurocognitive evaluations and oral periodontal examinations.

To ensure the integrity of the results, stringent exclusion criteria were implemented. Subjects were excluded if they were undergoing periodontal or orthodontic treatment or had received such treatment within the preceding three months were at risk for endocarditis due to periodontal probing, or had received antibiotic treatment within the previous three months. Participants with a clinical diagnosis of dementia, including Parkinson’s or Alzheimer’s disease, were also excluded. Based on these criteria, a total of 30 participants were included in the final study cohort. This timeframe of three months was chosen to minimize confounding effects from acute medical conditions, recent systemic infections, or dental treatments that could influence both periodontal health and cognitive assessments.

The study protocol received approval from the Ethics Committee of the Universidad Nacional de Colombia (Act No. 11-23 of 2023), ensuring adherence to the ethical principles delineated in the Declaration of Helsinki by the World Medical Association. Informed consent documents were developed to ensure transparency and comprehensive disclosure to potential participants. These documents provided detailed information regarding the study’s objectives, methodology, potential risks, and anticipated benefits, enabling participants to make informed decisions concerning their involvement. All participants provided written consent, affirming their voluntary participation and comprehension of the study’s procedures and ethical safeguards. Of note, data privacy measures were rigorously implemented in this study. All collected data were anonymized and stored securely in digital databases. Data sharing adhered to institutional and international ethical guidelines, ensuring that only de-identified datasets were shared for collaborative research purposes and solely with explicit approvals.

### 2.2. Neurocognitive Assessment

The study participants underwent comprehensive evaluations conducted by clinical professionals from the Neurosciences Department at the Universidad Nacional de Colombia. Detailed clinical histories were obtained. Cognitive function was assessed utilizing the Montreal Cognitive Assessment (MoCA) test, a validated instrument developed by Nasreddine et al. [27]. MoCA is widely recognized as an appropriate tool for identifying MCI [28]. It was specifically designed to detect MCI and has been shown to have greater predictive accuracy compared to other cognitive assessments like the Mini-Mental State Examination (MMSE) [29]. MoCA has been recently reported to exhibit a sensitivity of 73.5% to 83.8% and a specificity of 70.8% to 91.3% for detecting MCI, contingent upon the utilized cutoff scores [30]. This assessment instrument evaluates multiple cognitive domains, encompassing attention, concentration, executive function (including abstraction), memory, language, visuoconstructive abilities, calculation, and orientation [30]. Based on the score ranges in the MoCa test recently validated for MCI [31,32], this study categorizes cognitive function as follows: scores of 26 or higher indicate normal cognition, scores between 18 and 25 indicate MCI, scores between 10 and 17 indicate moderate cognitive impairment, and scores below 10 points indicate severe cognitive impairment.

### 2.3. Determination of the Educational Level

Participants’ educational levels were systematically recorded and categorized into three groups to investigate potential associations with MCI, given that education is a critical factor in assessing predisposition to developing MCI, as indicated by previous research [33]. Basic education referred to individuals who had completed primary school but had not progressed beyond secondary education, intermediate education encompassed participants who had successfully completed secondary education without pursuing tertiary education, and advanced education comprised those who had obtained a university degree, either at the undergraduate or postgraduate level, as well as individuals with technical certifications.

### 2.4. Periodontal Examination

A comprehensive dental examination was conducted by the faculty and residents of the Periodontics Postgraduate Program at the Faculty of Dentistry, Universidad Nacional de Colombia. All examiners underwent calibration training using the Polson criteria to ensure consistency and accuracy in their assessments [34]. The periodontal parameters evaluated comprised the following: (1) number of teeth present; (2) gingival erythema (inter-examiner Kappa Index [IE], 0.654); (3) pocket depth (PD) (IE, 0.609); (4) gingival margin position (IE, 0.432); (5) bleeding on probing (BoP) (IE, 0.583); and (6) plaque index (PI) (as described by O’Leary IE, 0.543). These parameters were systematically measured on six surfaces of each tooth, including distal vestibular [DV], vestibular [V], mesial vestibular [MV], distal lingual [DL], lingual [L], and mesial lingual [ML], excluding teeth restored with implants and third molars. A University of North Carolina probe was employed for all measurements to ensure precision and reproducibility. The rigorous calibration process, based on the Polson criteria, included training that encompassed theoretical sessions on periodontal parameters and practical exercises utilizing the University of North Carolina probe to measure PD, gingival margin position, and PI. Inter-examiner reliability was evaluated using Kappa Index values, demonstrating moderate to substantial agreement (e.g., PD = 0.609, PI = 0.543). Periodic recalibration and feedback addressed variability, ensuring standardized application of CDC/AAP guidelines for periodontitis diagnosis.

The diagnosis of periodontitis was established based on two internationally recognized classification systems. According to the Centers for Disease Control and Prevention/American Academy of Periodontology (CDC/AAP) guidelines [35], participants were categorized based on clinical attachment loss (CAL) and PD. Mild periodontitis was characterized by the presence of at least two interproximal surfaces with a CAL of three millimeters or more, at least two interproximal surfaces with a PD of four millimeters or more (on different teeth), or one surface with a PD of five millimeters or more. Moderate periodontitis was defined by at least two interproximal surfaces with a CAL of four millimeters or more (on different teeth) or two interproximal surfaces with a PD of five millimeters or more. Severe periodontitis was diagnosed when at least two interproximal surfaces exhibited a CAL of six millimeters or more (on different teeth), and at least one interproximal surface presented a PD of five millimeters or more [36].

According to the classification established by the American Academy of Periodontology in association with the European Federation of Periodontology during the 2017 World Workshop (AAP/EFP) [37,38], periodontitis was diagnosed when interproximal CAL was observed in at least two non-adjacent teeth or when vestibular or lingual CAL of three millimeters or more was accompanied by a PD exceeding three millimeters in at least two teeth [37]. This system further categorized periodontal health as exhibiting less than 10% BoP, with PD of three millimeters or less, and no interdental CAL or gingivitis. Gingivitis was defined as BoP of 10% or more, with PD of three millimeters or less, and no interdental CAL. Periodontitis stages ranged from stage I, characterized by interdental CAL of one to two millimeters, to stage IV, which involved CAL of five millimeters or more, with substantial tooth loss attributed to periodontitis [37].

### 2.5. Statistical Analyses

The Shapiro–Wilk test was employed to assess the normality of data distributions across the study groups. Clinical and sociodemographic variables related to cognitive function were analyzed using the Chi-squared (Chi^2^) test and Fisher’s exact test, as appropriate. For the analysis of periodontal parameters based on cognitive function, two statistical approaches were implemented depending on the distribution of the data. For non-normally distributed data, the Mann–Whitney U test was utilized to compare median values of periodontal parameters. Conversely, for normally distributed data, the Student’s T-test was applied.

Multiple linear regression analysis was performed using R software (version 4.4.0) to identify the most significant periodontal parameters for inclusion in the predictive model. The selection process involved evaluating various combinations of periodontal parameters and their contributions to predicting cognitive outcomes. The selected parameters were subsequently incorporated into a cluster prediction model analysis using MassHunter Mass Profiler Professional Software (Agilent Technologies, Santa Clara, CA, USA). The Partial Least Squares Discriminant (PLSD) model was applied for its suitability in analyzing complex, high-dimensional data and its effectiveness in distinguishing between categorical outcomes, such as control and MCI groups [39].

Furthermore, considering that Naïve Bayes models have been extensively utilized for clinical variables modeling due to their simplicity and efficacy and that these models have demonstrated promising results in predicting patient outcomes, particularly in the context of Alzheimer’s disease [40], the Naïve Bayes model was employed as a complementary approach, given its parsimony and utility in classification tasks. All statistical analyses were conducted with a 95% confidence level using Stata version 14. A *p*-value of ≤0.05 was considered indicative of statistical significance for all tests performed [41,42].

## 3. Results

### 3.1. Gathering and Evaluation of the PerioMind Colombia Cohort

The recruitment and experimental design of the study are summarized in Figure 1. A total of 70 participants were initially assessed, but 40 were excluded based on predetermined criteria, leaving a final cohort of 30 participants. These individuals were classified into two groups according to their cognitive function: the Control group (n = 20) and the MCI group (n = 10), as detailed in Table 1. The mean age of the participants was 70 ± 5 years (Table 1). In the Control group, the median age (IQR) was 67 (66–72), while in the MCI group it was 69 (68–72) (Table 2). A notable demographic feature was the predominance of women over men, with a 2:1 ratio in both groups. Comprehensive demographic and cohort assessment data are provided in Table 1 and Table 2.

Neuropsychological and dental assessments suggested that participants in the MCI group were less likely to have an advanced educational level compared to the Control group (Table 2). Additionally, a lower median number of teeth was observed in the MCI group compared to the Control group, with similar patterns noted for the upper jaw. For the lower jaw, tooth counts appeared comparable between groups. While these differences reflect trends in the data, they did not reach statistical significance (Table 2).

### 3.2. Periodontal and Cognitive-Linked Parameters in the PerioMind Colombia Cohort

The analysis of periodontal parameters demonstrated significant differences between the Control and MCI groups. Individuals in the MCI group exhibited higher median values for gingival redness, PD at 4 mm, and PD≥ 4 mm (Table 3). Although not statistically significant, a trend was observed for total PD (Table 3). Interestingly, a higher proportion of participants in the Control group exhibited PI levels within the 0–20% range compared to the MCI group. This trend approached statistical significance, suggesting potential differences in oral hygiene habits or outcomes between the groups (Table 3).

### 3.3. Generation of Class-Predictive Models for Identifying Mild Cognitive Impairment Risk

To develop a screening tool for identifying MCI risk based on periodontal parameters, predictive models were constructed using data from the PerioMind Colombia Cohort. The Partial Least Squares Discriminant (PLSD) and Naïve Bayes models were applied to analyze the selected periodontal parameters and generate class-predictive models (Table 4). These models aim to assist dental practitioners and oral health professionals in identifying individuals at risk of MCI. The models demonstrated high accuracy, correctly classifying participants into either the control or MCI group, with 100% accuracy in the Naïve Bayes model and 96.7% accuracy in the PLSD model (Appendix A). These results highlight the robustness of these methods in correctly classifying outcomes, supporting their applicability in predictive analyses. Among the periodontal parameters, gingival redness and PD ≥ 4 mm were the most effective at distinguishing individuals with MCI. Specifically, individuals with pronounced gingival redness and PD ≥ 4 mm greater than 0.3 exhibited 3.15 and 2.59 times greater odds of belonging to the MCI group than age-matched controls, respectively (Table 4). The model parameters exhibited positive interactions, with gingival redness showing the strongest predictive interaction (18.73%), followed by periodontitis according to the CDC/AAP classification (14.07%), moderate or severe periodontitis (6.13%), and PD ≥ 4 mm (4.54%) (Table 4). Notably, all participants classified as having periodontitis based on the CDC/AAP criteria, as well as those in advanced periodontal severity stages (stage III or IV in the EFP/AAP classification and moderate or severe periodontitis in the CDC/AAP classification), were consistently categorized in the MCI group. Lower educational attainment was also strongly associated with a higher probability of being classified in the MCI group. Comprehensive predictive performance metrics for both the Naïve Bayes and PLSD models are detailed in Appendix A.

## 4. Discussion

Epidemiological studies suggest that conditions such as periodontitis, tooth loss, and other oral diseases may contribute to MCI [43], potentially through mechanisms involving systemic inflammation and neuroinflammatory processes [44]. Recent findings indicate a bidirectional relationship between oral health and cognitive dysfunction, emphasizing the potential for poor oral health to exacerbate cognitive impairment and vice versa [43]. Of note, a recent systematic review and meta-analysis found a significant association between periodontitis and an increased risk of MCI, with a pooled odds ratio of 1.70, indicating that individuals with periodontitis are more likely to develop MCI compared to those without the condition [45].

Our research contributes to those above, expanding the corpus of evidence from a novel perspective by focusing on specific periodontal parameters and their associations with MCI, an area that has not been previously explored in depth to the best of our knowledge. Through the evaluation of periodontal indices in the context of MCI risk within a diverse elderly population, we provide new insights into the role of oral health in cognitive decline. This research adds a layer of specificity to the understanding of how periodontal health may influence cognitive function, offering potential new avenues for risk assessment and early diagnosis of MCI.

The data collected here indicates that subjects diagnosed with MCI had a lower average number of teeth compared to those with normal cognitive function, particularly in the upper jaw. This aligns with previous studies showing that participants with fewer than 20 teeth exhibited reduced cognitive capacity [46] and that partial or total tooth loss is associated with lower cognitive scores [47]. Research has also highlighted that chewing with natural teeth supports brain activity, cerebral blood flow, and hippocampal and cortical function, while tooth loss reduces sensory input and alters neurotransmission [48,49]. Additionally, individuals with fewer teeth, particularly edentulous individuals, have an increased risk of cognitive impairment [50], as observed in older adults with fewer than 10 teeth performing worse on cognitive assessments [51]. These findings underscore the importance of preserving natural teeth and addressing tooth loss as part of strategies to mitigate cognitive decline. Further research is needed to explore factors such as dental prosthesis use and its potential impact on cognitive health [52].

Similarly, educational level is a well-recognized determinant of cognitive reserve and an essential factor influencing the onset and progression of MCI [53]. Higher educational attainment is often associated with enhanced cognitive resilience [53], enabling individuals to better compensate for neuropathological changes. Conversely, lower educational levels have been linked to increased vulnerability to MCI [54], often compounded by socioeconomic disparities, limited access to healthcare, and lifestyle factors. Despite its significance, educational level frequently acts as a confounding variable in studies investigating MCI, complicating efforts to isolate the influence of other risk factors [55], such as health comorbidities, including periodontal disease.

Our study, conducted within the PerioMind Colombia Cohort, highlights the critical interplay between educational level and periodontal disease in predicting MCI. Participants with stage IV periodontitis were more likely to have basic education, consistent with national surveys reporting a higher prevalence of severe periodontitis among individuals with lower educational levels [35]. Limited education may contribute to poorer oral health literacy and access to preventive care, compounding the risk of both periodontal disease and cognitive decline [56]. By incorporating educational attainment into predictive models, we identified specific periodontal metrics, such as gingival redness and severe periodontitis, strongly associated with MCI in individuals with lower education levels. These findings underscore that cognitive reserve, influenced by educational attainment, may modulate the impact of periodontal disease on cognitive outcomes. Notably, our results indicate that routine periodontal examinations, in conjunction with an understanding of educational disparities, could function as cost-effective mechanisms for identifying individuals at risk of MCI, thereby facilitating targeted early interventions and comprehensive public health strategies.

In a related context, one of the most significant associations identified in this study was between increased gingival erythema and MCI, rendering it the most prominent factor associated with cognitive decline in our cohort. Previous research corroborates the role of gingival inflammation, rather than hemorrhage, as a significant predictor of cognitive impairment [57]. This relationship may be influenced by advanced age and underlying cardiovascular conditions, which are known to contribute to systemic inflammation, neurodegeneration, and potentially gingival health [58,59].

Here, we also provide significant thresholds for periodontal parameters, such as gingival erythema and PD, derived from class-predictive models that demonstrate a substantial correlation with the risk of MCI in individuals over 70 years of age. These parameters function as practical indicators during routine periodontal examinations, offering a methodology to stratify patients at risk of dementia in clinical practice. The predictive models developed in this research exhibited potential for preliminary screening purposes. However, they necessitate further validation and testing in larger and more diverse populations to confirm their reliability and applicability. Expanding these efforts will enhance their utility and contribute to effective early detection and intervention strategies, ultimately improving outcomes for older adults globally [9,41,60,61,62].

It is also crucial to emphasize that in both developed and developing nations, dental clinics frequently serve as accessible healthcare points for older adults, providing routine and preventive treatments. This accessibility offers an opportunity to integrate periodontal assessments into broader healthcare strategies to identify individuals at risk for cognitive decline. Research has demonstrated the feasibility of linking oral health services with systemic healthcare in regions such as Nigeria, Brazil, and North America [63,64,65]. Nevertheless, neuropsychological evaluations remain less accessible due to their complexity, associated costs, and limited availability of trained professionals and diagnostic instruments [66]. Enabling dental professionals to identify patients who may benefit from specialized care could address this disparity. Early detection of individuals with MCI or early dementia stages facilitates timely intervention, potentially decelerating cognitive decline, enhancing quality of life, and delaying the progression to severe dementia [59,67].

## 5. Conclusions

This study introduces a novel approach to clinically leverage the link between specific periodontal parameters and MCI. By examining clinically measurable indicators like gingival erythema and PD, we identify significant MCI-associated thresholds that can be applied in routine periodontal examinations. Consequently, based on the findings elucidated herein, individuals within the predicted MCI risk thresholds should undergo neuropsychological evaluations to rule out potential MCI.

The data obtained also underscore the potential of oral health screenings to address cognitive healthcare disparities, especially in underserved populations. Predictive models emphasize the significance of interdisciplinary approaches, integrating dental and cognitive assessments for a comprehensive strategy to address cognitive decline. Consideration of educational attainment is strongly recommended, as elucidated in the findings. Further investigations to validate and expand these results could substantially enhance early intervention practices and improve outcomes for elderly individuals at risk of age-related dementia globally.

## 6. Limitations of the Study

The authors acknowledge that the relatively small sample size, despite extensive recruitment efforts and comprehensive neuropsychological and periodontal assessments, could be considered a limitation of this study. While the results provide valuable proof of concept, further analysis in larger cohorts is needed to validate the findings. Nevertheless, it is important to highlight that the number of participants included was sufficient to allow for meaningful comparisons of various periodontal parameters with an acceptable level of statistical confidence. Furthermore, we acknowledge the limitations of the MoCA in populations with low literacy levels. To address this constraint, we incorporated specific measurements of literacy levels and educational attainment in our cohort. These factors were subsequently integrated into our analysis to account for their potential influence on cognitive outcomes, thereby ensuring a more robust and context-sensitive evaluation.

It is also imperative to acknowledge that lifestyle factors and healthcare accessibility may exert significant influence on both periodontitis and cognitive outcomes. Subsequent investigations should take into account the variability of these factors in the relationship between periodontitis and MCI, as elucidated by the findings presented in this study.

Additionally, because most participants were examined in an institution lacking access to comprehensive dental assessment equipment, intraoral radiographs could not be obtained. However, the interdental CAL considered in this investigation is the gold standard for determining the severity of periodontitis. It allows for the placement of the patient into Stage I to IV, as discussed in the online Journal of Clinical Periodontology within the framework of the AAP/EFP16. X-rays are also unnecessary for appropriate classification according to the CDC/AAP case definition [35]. This approach ensured that the evaluation remained robust and clinically relevant.

## Figures and Tables

**Figure 1 biomedicines-13-00205-f001:**
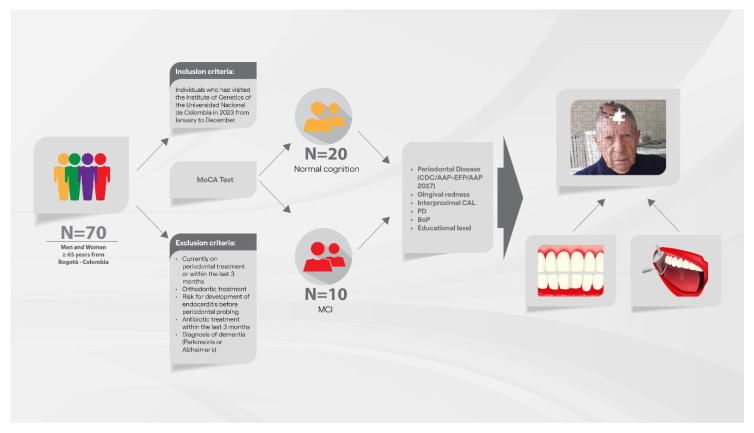
Workflow and main findings of the study. The diagram illustrates the comprehensive workflow of the study, including cohort selection, experimental design, data acquisition through periodontal and neuropsychological assessments, data analysis, and the clinical relevance of periodontal screening in identifying associations with unhealthy cognitive aging.

**Table 1 biomedicines-13-00205-t001:** Descriptive analysis of socio-demographic and clinical variables of the PerioMind Colombia Cohort. The mean ± standard deviation (SD) is provided for age, while absolute frequencies (n) and percentages (%) are used for categorical variables such as sex, educational level, and cognitive function. The cohort includes 30 elderly individuals stratified by cognitive function into control and mild cognitive impairment (MCI) groups.

Variable		Mean/Frequency	Percentage (%)
Age		70 ± 5	-
Sex	Female	20	66.7
	Male	10	33.3
Educational level	Basic	11	36.7
	Intermediate	12	40.0
	Advanced	7	23.3
Cognitive function	Control	20	66.7
	Mild Cognitive Impairment (MCI)	10	33.3

**Table 2 biomedicines-13-00205-t002:** Clinical and sociodemographic variables of participants of the PerioMind Colombian Cohort grouped into Control and MCI categories. The data include median values and interquartile ranges (IQR) for age and number of teeth (overall, upper jaw, and lower jaw), along with the distribution of sex and educational levels across groups.

		Control		MCI	
		n = 20	Median (IQR)	n = 10	Median (IQR)
Age		-	67 (66–72)	-	69 (68–72)
Sex	Female	13	-	7	-
	Male	7	-	3	-
Educational level	Basic	7	-	4	-
	Intermediate	7	-	5	-
	Advanced	6	-	1	-
Number of teeth		-	22 (16–25)	-	20 (16–27)
Number of teeth (Upper Jaw)		-	10 (7–12.5)	-	9 (5–12)
Number of teeth (Lower Jaw)		-	11 (9.5–13.5)	-	11 (9.5–13.5)

**Table 3 biomedicines-13-00205-t003:** Comparison of periodontal clinical variables based on cognitive function between Control and mild cognitive impairment (MCI) groups. Values for median (IQR) and mean ± SD are provided, along with corresponding *p*-values for statistical analyses. Significant differences (*p* < 0.05) are marked with †. Medians were compared using the Mann–Whitney U test, Chi^2^ test, and Fisher’s exact test, whereas means were compared using the *t*-test. Abbreviations: MoCA, Montreal Cognitive Assessment; PI, plaque index; BoP, bleeding on probing; PD, pocket depth; CAL, clinical attachment loss.

		Cognitive Function		*p* Value
		Control	MCI	
MoCA Test	Median (IQR)	27 (26–28)	21 (21–23)	^†^ 0.000
PI	Mean ± SD	0.4 ± 0.31	0.4 ± 0.1	0.996
PI	Median (IQR) 0–20%	n = 6 (100%)	n = 0 (0%)	0.053
	Median (IQR) > 20%	n = 14 (58.33%)	n = 10 (41.67%)	
Gingival redness	Median (IQR)	0.0 (0.0–0.16)	0.39 (0.31–0.59)	^†^ 0.003
BoP	Mean ± SD	0.34 ± 0.22	0.31 ± 0.15	0.672
Total PD	Median (IQR)	2.19 (1.88–2.5)	2.51 (2.22–2.91)	0.074
PD ≥ 3 mm	Mean ± SD	0.38 ± 0.24	0.51 ± 0.16	0.143
PD = 4 mm	Median (IQR)	0.03 (0.02–0.07)	0.1 (0.05–0.18)	^†^ 0.015
PD ≥4 mm	Median (IQR)	0.05 (0.02–0.1)	0.18 (0.05–0.2)	^†^ 0.044
PD = 5 mm	Median (IQR)	0.01 (0.0–0.02)	0.04 (0.0–0.06)	0.169
PD ≥ 5 mm	Median (IQR)	0.01 (0.0–0.04)	0.05 (0.0–0.12)	0.350
PD ≥ 6 mm	Median (IQR)	0.0 (0.0–0.02)	0.01 (0.0–0.04)	0.713
Interproximal CAL 1/2 mm	Mean ± SD	0.4 ± 0.2	0.43 ± 0.2	0.687
Interproximal CAL 3/4 mm	Mean ± SD	0.32 ± 0.18	0.27 ± 0.09	0.352
Interproximal CAL ≥ 5 mm	Median (IQR)	0.1 (0.04–0.15)	0.12 (0.1–0.2)	0.350
Interproximal CAL ≥ 6 mm	Median (IQR)	0.03 (0–0.07)	0.05 (0.03–0.09)	0.169

**Table 4 biomedicines-13-00205-t004:** Results of predictive models for identifying MCI risk based on periodontal and educational parameters applied to the PerioMind Colombia Cohort. Parameter ranges for Control and mild cognitive impairment (MCI) groups, the ratio of MCI to Control, interaction percentages, and interaction directions are detailed. Predictive values were generated using Naïve Bayes and Partial Least Squares Discriminant (PLSD) models. * Strong positive interactions are denoted by “p”. Abbreviations: CDC/AAP, Centers for Disease Control and Prevention in association with the American Academy of Periodontology; EFP/AAP, European Federation of Periodontology in association with the American Academy of Periodontology.

	Range Control	Range MCI	Ratio MCI/Control	Interaction (%)	Interaction Direction *
Education level	2.37–2.77	<2.37	0.87	-	-
Stages Periodontal disease (EFP/AAP)	2.65–3.08	>3.08	1.06	-	-
Stages Periodontal disease (CDC/AAP)	1.34–1.97	>1.97	1.11	6.13	p
Periodontal Disease (CDC/AAP)	≤1	1	1.11	14.07	p
Gingival redness	0–0.281	>0.281	3.15	18.73	p
PD ≥ 4 mm	0–0.30	>0.3	2.59	4.54	p

## Data Availability

All raw clinical data generated in this study from the PerioMind Colombia Cohort has been made publicly available through the clinical data repository https://data.mendeley.com/datasets/c3jk6nwr56/1 (accessed on 19 December 2024) under the identifier: Arévalo Caro, Catalina María; López, Diego; Sánchez Milán, Jose Antonio; Lorca, Cristina; Mulet, María; Arboleda, Humberto; Losada, Sergio; Gallart-Palau, Xavier; Serra, Aida (2024), “Anonymised PerioMind Colombia cohort database”, Mendeley Data, V1, doi: 10.17632/c3jk6nwr56.1. This ensures the open and prospective use of the data by the scientific community in accordance with the recommendations outlined in the research group’s publication [60].

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
