# Peer review of "Periodontal Indices as Predictors of Cognitive Decline: Insights from the PerioMind Colombia Cohort"

_biomedicines, 2025, doi:10.3390/biomedicines13010205_

Round 1

Reviewer 1 Report

Comments and Suggestions for Authors

For the Introduction:

1. Introduction is too general. It talks about periodontitis broadly, but not specifically about the study. Reorganize it to highlight why this research is important.

2. There are some texts that repeats information about periodontitis too often. It discusses its impact on health in similar ways multiple times. Cut out any sentences that do not add new information.

3. The introduction does not explain why certain methods were used. Why were periodontal and cognitive tests chosen? Add a short explanation for these choices. Readers need to know why these methods fit the research.

4. Socioeconomic issues are mentioned briefly but not explained. Explain why these issues matter in the study.

5. The introduction does not clearly connect periodontitis to cognitive decline. Use examples or studies to support this connection. Add more explanation about how these two are related.

6. Explain why the Colombian population is a good focus. Contrast global and local findings to make this clear.

For the Related Literature:

7. Some cited studies are old. Replace them with newer studies where possible.

8. The review lists studies without much analysis. Explain what each study adds to the research. Highlight gaps or limitations in the literature.

9. Some studies are explained well, others are just mentioned. Discuss each study equally, especially key ones.

10. The review ignores variables like lifestyle or healthcare access. These could influence both periodontitis and cognition. Readers need to know what factors might complicate the results.

For the Methodology, Results and Discussions:

11. The study uses a cross-sectional design. It is not explained why this design was chosen. Explain why cross-sectional design is better for this study.

12. The study lists strict exclusion criteria. A three-month period is set for excluding certain participants. There is no explanation for this specific time frame.

13. MoCA has known limitations in low-literacy populations. Include additional tools or methods to address this limitation, or justification.

14. Both Mann–Whitney U and T-tests are used. The sample size is very small (n=30). Consider how small size affects the strength of conclusions.

15. The study follows strong ethical guidelines. However, data privacy measures are not described. Explain storage, sharing, and access restrictions.

Author Response

Reviewer #1

For the Introduction:

  1. Introduction is too general. It talks about periodontitis broadly, but not specifically about the study. Reorganize it to highlight why this research is important.

RESPONSE: We thank Reviewer #1 for their thoughtful suggestion regarding the revision of the Introduction section. In response, we have completely rewritten the Introduction in the revised manuscript, following the advice provided by both the Academic Editor and Reviewer #1. The revised section adheres to the recommended guidelines and presents a well-rounded and structured introduction in several balanced paragraphs.

The revised Introduction begins with a broad overview, highlighting the global impact of dementia and its worldwide effects. It contextualizes the relevance of studying this issue specifically in Colombia and Latin America, emphasizing their unique demographic and epidemiological characteristics. Additionally, the revision highlights the importance and novelty of linking periodontal evaluations to the presence of MCI. The section concludes with a clear and concise statement of the study’s objectives, as recommended.

  1. There are some texts that repeats information about periodontitis too often. It discusses its impact on health in similar ways multiple times. Cut out any sentences that do not add new information.

RESPONSE: We thank Reviewer #1 for the thoughtful suggestion. In response, these paragraphs, along with most sections of the manuscript, have been rewritten in the revised version to eliminate redundancy and enhance clarity.

  1. The introduction does not explain why certain methods were used. Why were periodontal and cognitive tests chosen? Add a short explanation for these choices. Readers need to know why these methods fit the research.

RESPONSE: We thank Reviewer #1 for this suggestion. Following previous advice from the Academic Editor and this suggestion, we have ensured that these choices are properly justified in the revised Methods section, as follows: ‘Cognitive function was assessed utilizing the Montreal Cognitive Assessment (MoCA) test, a validated instrument developed by Nasreddine et al. [27]. MoCA is widely rec-ognized as an appropriate tool for identifying MCI [28]. It was specifically designed to detect MCI and has been shown to have greater predictive accuracy compared to other cognitive assessments like the Mini-Mental State Examination (MMSE) [29]. MoCA has been recently reported to exhibit a sensitivity of 73.5% to 83.8% and specificity of 70.8% to 91.3% for detecting MCI, contingent upon the utilized cutoff scores [30]. This assessment instrument evaluates multiple cognitive domains, encompassing attention, concentration, executive function (including abstraction), memory, language, visuoconstructive abilities, calculation, and orientation [30]. Based on the score ranges in the MoCa test recently validated for MCI [31,32], this study categorizes cognitive function as follows: scores of 26 or higher indicate normal cognition, scores between 18 and 25 indicate MCI, scores between 10 and 17 indicate moderate cognitive impairment, and scores below 10 points indicate severe cognitive impairment.’ ‘The diagnosis of periodontitis was established based on two internationally recog-nized classification systems. According to the Centers for Disease Control and Preven-tion/American Academy of Periodontology (CDC/AAP) guidelines [35], participants were categorized based on clinical attachment loss (CAL) and pocket depth (PD). Mild peri-odontitis was characterized by the presence of at least two interproximal surfaces with CAL of three millimeters or more, at least two interproximal surfaces with PD of four millimeters or more (on different teeth), or one surface with a PD of five millimeters or more. Moderate periodontitis was defined by at least two interproximal surfaces with CAL of four millimeters or more (on different teeth) or two interproximal surfaces with PD of five millimeters or more. Severe periodontitis was diagnosed when at least two inter-proximal surfaces exhibited CAL of six millimeters or more (on different teeth), and at least one interproximal surface presented a PD of five millimeters or more [36].’

  1. Socioeconomic issues are mentioned briefly but not explained. Explain why these issues matter in the study.

RESPONSE: Following this suggestion from Reviewer #1, we have ensured that the socioeconomic particulars of the Latin American region and Colombia are properly detailed in the Introduction section of the revised manuscript, as follows: ‘In Latin America, the prevalence of aging-associated dementias is estimated to range between 5% and 10% among individuals over 60 years of age, with projections indicating a significant upward trend consistent with global patterns [3-5]. Recent studies emphasize the significance of conducting research within this specific elderly population, as mis-diagnosis and underdiagnosis remain prevalent [6]. These challenges are primarily at-tributed to limited access to culturally appropriate diagnostic tools and persistent healthcare disparities across countries in the region [7], as well as the influence of the mestizo population's genetic diversity, which complicates dementia characterization [7]. Similarly, most research is based on European ancestry, potentially leading to biases in understanding and treating dementia in mestizo populations [7]. Colombia presents an optimal setting for investigating mestizo populations within Latin America owing to its heterogeneous genetic and cultural heritage, which exemplifies the broader demographic patterns of the region [7]. Furthermore, Colombia's increasing emphasis on public health research, in conjunction with its healthcare challenges, renders it a significant context for examining global health disparities and population dynamics [6]. Dementia prevalence in Colombia ranges from 5% to 7% in individuals over 60 years of age, aligning with the aforementioned broader Latin American trends [6]. However, specific geographical areas within the country have exhibited an exceptionally high prevalence of 23.6% in individuals over 60 years old, attributed to socioeconomic factors and poorly defined chronic comorbidities [8].’

  1. The introduction does not clearly connect periodontitis to cognitive decline. Use examples or studies to support this connection. Add more explanation about how these two are related.

RESPONSE: We thank Reviewer #1 for pointing out this limitation in the Introduction. We have now further rewritten the entire Introduction section to clearly support the connection of these crucial concepts to the aim of the manuscript in the revised submission, as follows: ‘Notably, the deleterious effects of periodontitis extend beyond the oral cavity and have been associated with systemic health issues, as we and other researchers have previously elucidated [19-21].  Furthermore, this disease appears to be a chronic comorbidity commonly observed in individuals affected by aging-related dementias [22]. Emerging epidemiological data from various countries suggest that periodontitis may play a significant role in influencing the onset and progression of age-related dementias [23]. This association is likely attributable to the systemic inflammation and immune responses elicited by chronic periodontal infections, which could exacerbate neuroinflammatory processes linked to cognitive decline [23]. Additionally, the prevalence of periodontitis in aging populations underscores its potential as a modifiable risk factor for dementia [22]. Elucidating this connection could facilitate the development of targeted preventive strategies to mitigate dementia's burden in susceptible populations.

  1. Explain why the Colombian population is a good focus. Contrast global and local findings to make this clear.

RESPONSE: We thank Reviewer #1 for highlighting this important focus to add to the Introduction. Following the advice, we have now justified the manuscript's focus on the Colombian population in the revised Introduction section, as follows: ‘Colombia presents an optimal setting for investigating mestizo populations within Latin America owing to its heterogeneous genetic and cultural heritage, which exemplifies the broader demographic patterns of the region [7]. Furthermore, Colombia's increasing emphasis on public health research, in conjunction with its healthcare challenges, renders it a significant context for examining global health disparities and population dynamics [6]. Dementia prevalence in Colombia ranges from 5% to 7% in individuals over 60 years of age, aligning with the aforementioned broader Latin American trends [6]. However, specific geographical areas within the country have exhibited an exceptionally high prevalence of 23.6% in individuals over 60 years old, attributed to socioeconomic factors and poorly defined chronic comorbidities [8].’

For the Related Literature:

  1. Some cited studies are old. Replace them with newer studies where possible.

RESPONSE: We thank Reviewer #1 for highlighting this point. In response, we have rewritten the Introduction section and included updated references to better justify the study, as suggested by Reviewer #1 and the Academic Editor.

  1. The review lists studies without much analysis. Explain what each study adds to the research. Highlight gaps or limitations in the literature.

RESPONSE: We thank Reviewer #1 for highlighting this point. Following this advice, we have ensured that the cited literature is better contextualized in the revised manuscript.

  1. Some studies are explained well, others are just mentioned. Discuss each study equally, especially key ones.

RESPONSE: In response to Reviewer #1's valuable observation, we have diligently incorporated their suggestion by enhancing the contextualization and discussion of the cited literature throughout the relevant sections of the revised manuscript.

  1. The review ignores variables like lifestyle or healthcare access. These could influence both periodontitis and cognition. Readers need to know what factors might complicate the results.

RESPONSE: In acknowledgment of Reviewer #1's relevant observation regarding this critical issue, the authors have incorporated the identified limitation into the corresponding section of the revised manuscript. Specifically, the Limitations section now includes the following addition: ‘It is also imperative to acknowledge that lifestyle factors and healthcare accessibility may exert significant influence on both periodontitis and cognitive outcomes. Subsequent investigations should take into account the variability of these factors in the relationship between periodontitis and MCI, as elucidated by the findings presented in this study.’

For the Methodology, Results and Discussions:

  1. The study uses a cross-sectional design. It is not explained why this design was chosen. Explain why cross-sectional design is better for this study.

RESPONSE: The authors express gratitude to Reviewer #1 for this recommendation. As indicated, the justification for employing this methodological design has been incorporated into the revised manuscript, as follows: A cross-sectional observational study was designed and conducted at the Institute of Genetics, Universidad Nacional de Colombia. This research design was selected due to its suitability for exploratory investigations aimed at identifying specific clinical associations among variables, as previously delineated [24]. This approach allows for the efficient collection of representative data within resource constraints, providing preliminary ev-idence to guide future longitudinal studies for causal confirmation [24].

  1. The study lists strict exclusion criteria. A three-month period is set for excluding certain participants. There is no explanation for this specific time frame.

RESPONSE: We appreciate the reviewer's observation regarding the three-month exclusion period. In response, we have specifically justified this criterion in the revised Methodology section of the manuscript as follows: ‘This timeframe was chosen to minimize confounding effects from acute medical conditions, recent systemic infections, or dental treatments that could influence both periodontal health and cognitive assessments.

  1. MoCA has known limitations in low-literacy populations. Include additional tools or methods to address this limitation, or justification.

RESPONSE: We thank Reviewer #1 for highlighting this point. In response, we have added this limitation to the Limitations section of the revised manuscript, as follows: ‘Furthermore, we acknowledge the limitations of the MoCA in populations with low literacy levels. To address this constraint, we incorporated specific measurements of literacy levels and educational attainment in our cohort. These factors were subsequently integrated into our analysis to account for their potential influence on cognitive outcomes, thereby ensuring a more robust and context-sensitive evaluation.’

  1. Both Mann–Whitney U and T-tests are used. The sample size is very small (n=30). Consider how small size affects the strength of conclusions.

RESPONSE: The authors express gratitude to Reviewer #1 for drawing attention to the utilization of both Mann–Whitney U and T-tests in relation to the available sample size. This limitation, as per the Reviewer's recommendation, has been acknowledged in the revised manuscript's Limitations section, emphasizing the necessity for larger-scale studies to validate the significant findings initially reported in this investigation.

  1. The study follows strong ethical guidelines. However, data privacy measures are not described. Explain storage, sharing, and access restrictions.

RESPONSE: We express our gratitude to Reviewer #1 for drawing attention to this significant aspect. Following that suggestion, we have now clarified at the conclusion of the initial sub-heading of the revised methods section that: 'It is noteworthy that stringent data privacy measures were implemented in this study. All collected data were anonymized and stored securely in encrypted digital databases with access restricted to authorized research team members. Data sharing adhered to institutional and international ethical guidelines, ensuring that only de-identified datasets were shared for collaborative research purposes, and exclusively with explicit approvals.'

We finally and sincerely thank Reviewer #1 for the thoughtful consideration of our work, the time spent reviewing our manuscript, and the valuable suggestions and feedback provided. We strongly believe that these contributions have undoubtedly enhanced the rigor and scientific quality of the revised manuscript.

Reviewer 2 Report

Comments and Suggestions for Authors

Dear authors,

thank you for your intriguing manuscript. Here are my thoughts on how to improve its quality even more:

1) Introduction:

- Describe why Colombia's mestizo population was selected for this study and how it relates to health disparities around the world.

2) Methods:

- Give more evidence to support the small sample size and any possible drawbacks.

- Describe in more detail the examiners' calibration procedure and its importance for preserving uniformity.

3) Results:

- To prevent table duplication, concentrate on the narrative's most statistically significant findings.

- Emphasize in the text how accurate the PLSD and Naïve Bayes models are at making predictions.

4) Discussion

- Expand on the practical implications for clinicians, particularly dentists working in low-resource environments. 

- Examine the possibility of integrating periodontal evaluations into more comprehensive public health plans.

5) In conclusion, highlight practical suggestions for incorporating periodontal screening into regular dental checkups, especially for senior citizens.

Warm regards!

Comments on the Quality of English Language

Simplify long sentences and make sure that phrases like "cognitive decline" and "periodontal parameters" are used consistently. Example: Replace "empirical evidence suggests that periodontal therapy can lead to clinical improvement in patients with MCI" with "Evidence suggests that periodontal therapy improves clinical outcomes in MCI patients."

Author Response

Reviewer #2

Dear authors,

thank you for your intriguing manuscript. Here are my thoughts on how to improve its quality even more:

RESPONSE: The authors sincerely express their appreciation to Reviewer #2 for the thoughtful consideration of their work, for recognizing the novelty and clinical potential of the study, and for providing constructive feedback to further enhance the revised manuscript.

1) Introduction:

- Describe why Colombia's mestizo population was selected for this study and how it relates to health disparities around the world.

RESPONSE: We express our gratitude to Reviewer #2 for this valuable suggestion, which aligns with the revisions proposed by Reviewer #1 and the Academic Editor. Consequently, we have addressed this matter by comprehensively revising the introduction section in the updated manuscript, incorporating current references and specifically addressing this point as follows: ‘[…] Recent studies emphasize the significance of conducting research within this specific elderly population, as mis-diagnosis and underdiagnosis remain prevalent [6]. These challenges are primarily at-tributed to limited access to culturally appropriate diagnostic tools and persistent healthcare disparities across countries in the region [7], as well as the influence of the mestizo population's genetic diversity, which complicates dementia characterization [7]. Similarly, most research is based on European ancestry, potentially leading to biases in understanding and treating dementia in mestizo populations [7].

Colombia presents an optimal setting for investigating mestizo populations within Latin America owing to its heterogeneous genetic and cultural heritage, which exemplifies the broader demographic patterns of the region [7]. Furthermore, Colombia's increasing emphasis on public health research, in conjunction with its healthcare challenges, renders it a significant context for examining global health disparities and population dynamics [6]. Dementia prevalence in Colombia ranges from 5% to 7% in individuals over 60 years of age, aligning with the aforementioned broader Latin American trends [6]. However, specific geographical areas within the country have exhibited an exceptionally high prevalence of 23.6% in individuals over 60 years old, attributed to socioeconomic factors and poorly defined chronic comorbidities [8].’

2) Methods:

- Give more evidence to support the small sample size and any possible drawbacks.

RESPONSE: We express our appreciation to Reviewer #2 for this recommendation, which aligns with the modifications also proposed by Reviewer #1. Accordingly, in response to this suggestion from both reviewers, we have now explicitly acknowledged in the revised manuscript's Limitations section, emphasizing the necessity for larger-scale studies to validate the significant findings initially reported in this investigation and that, despite the limited sample size available for the PerioMind cohort, it has yielded statistically relevant results.

- Describe in more detail the examiners' calibration procedure and its importance for preserving uniformity.

RESPONSE: We express our gratitude to Reviewer #2 for emphasizing the importance of expanding this aspect of the manuscript. Consequently, we have provided further details regarding the calibration process as follows: ‘[…] The rigorous calibration process based on Polson criteria was predicated on training that encompassed theoretical sessions on periodontal parameters and practical exercises utilizing the University of North Carolina probe to measure PD, gingival margin position, and PI. Measurements were conducted on six tooth surfaces (distal vestibular, vestibular, mesial vestibular, distal lingual, lingual, and mesial lingual). Inter-examiner reliability was evaluated using Kappa Index values, demonstrating moderate to substantial agreement (e.g., PD = 0.609, PI = 0.543). Periodic recalibration and feedback addressed variability, ensuring standardized application of CDC/AAP guidelines for periodontitis diagnosis.’

3) Results:

- To prevent table duplication, concentrate on the narrative's most statistically significant findings.

RESPONSE: We thank Reviewer #2 for this suggestion, which aligns with a previous recommendation from the Academic Editor. In response, we have rewritten parts of the Results section and limited the use of statistical terms to relevant items, such as tables, as suggested.

- Emphasize in the text how accurate the PLSD and Naïve Bayes models are at making predictions.

RESPONSE: In response to the constructive feedback provided by Reviewer #2, we have now emphasized in the revised Results section that:‘ […] These models aim to assist dental practitioners and oral health professionals in identifying individuals at risk for MCI. The models demonstrated high accuracy, correctly classifying participants into either the control or MCI group, with 100% accuracy in the Naïve Bayes model and 96.7% accuracy in the PLSD model (Supplementary Tables 1-3). These results highlight the robustness of these methods in correctly classifying outcomes, supporting their applicability in predictive analyses.’

4) Discussion

- Expand on the practical implications for clinicians, particularly dentists working in low-resource environments.

- Examine the possibility of integrating periodontal evaluations into more comprehensive public health plans.

5) In conclusion, highlight practical suggestions for incorporating periodontal screening into regular dental checkups, especially for senior citizens.

RESPONSE: The authors express gratitude to Reviewer #2 for these valuable suggestions. In response, the entire Discussion section has been revised to specifically address the practical implications for clinicians, particularly dentists operating in resource-limited environments, and the potential integration of periodontal evaluations into comprehensive public health plans. Furthermore, the Conclusion has been modified to emphasize practical recommendations for incorporating periodontal screening into routine dental examinations, especially for the elderly population. These revisions aim to underscore the significance of this study's approach and its relevance to clinical practice, aligning with the insightful feedback provided by Reviewer #2

Simplify long sentences and make sure that phrases like "cognitive decline" and "periodontal parameters" are used consistently. Example: Replace "empirical evidence suggests that periodontal therapy can lead to clinical improvement in patients with MCI" with "Evidence suggests that periodontal therapy improves clinical outcomes in MCI patients."

RESPONSE: The authors acknowledge Reviewer #2 for this constructive suggestion. In response, the manuscript has been revised to simplify complex sentences throughout and ensure consistent usage of terminology such as "cognitive decline" and "periodontal parameters." Furthermore, the referred statement has been rewritten, along with the entire paragraph, in the revised manuscript to address the feedback provided by both reviewers and the academic editor.

We finally and sincerely thank Reviewer #2 for the thoughtful consideration of our work, the time spent reviewing our manuscript, and the valuable suggestions and feedback provided. We strongly believe that these contributions have undoubtedly enhanced the rigor and scientific quality of the revised manuscript.

Round 2

Reviewer 1 Report

Comments and Suggestions for Authors

1. Just a minor revision, please fix sentence flow, improve clarity, and grammars.

Reviewer 2 Report

Comments and Suggestions for Authors

Dear Authors,

thank you for considering my comments and suggestions to enhanced the quality of the manuscript.

In my opinion, your manuscript should be considered for publication.

Congrats and best regards!